# Implementation and Effectiveness of Novel Therapeutic Substances for Advanced Malignant Melanoma in Saxony, Germany, 2010–2020—Cohort Study Based on Administrative Data

**DOI:** 10.3390/cancers13246150

**Published:** 2021-12-07

**Authors:** Thomas Datzmann, Jochen Schmitt, Saskia Fuhrmann, Martin Roessler, Friedegund Meier, Olaf Schoffer

**Affiliations:** 1National Center for Tumor Diseases, 01307 Dresden, Germany; jochen.schmitt@uniklinikum-dresden.de (J.S.); Friedegund.Meier@uniklinikum-dresden.de (F.M.); 2German Cancer Research Center (DKFZ), 69120 Heidelberg, Germany; 3Faculty of Medicine and University Hospital Carl Gustav Carus, Technische Universität Dresden, 01307 Dresden, Germany; 4Helmholtz-Zentrum Dresden-Rossendorf (HZDR), 01328 Dresden, Germany; 5Medizinische Fakultät Carl Gustav Carus, Center for Evidence-Based Healthcare, TU Dresden, 01307 Dresden, Germany; saskia.fuhrmann@uniklinikum-dresden.de (S.F.); martin.roessler@uniklinikum-dresden.de (M.R.); olaf.schoffer@uniklinikum-dresden.de (O.S.); 6Hospital Pharmacy, University Hospital Carl Gustav Carus, 01307 Dresden, Germany; 7Medizinische Fakultät Carl Gustav Carus, Skin Cancer Center at the University Cancer Centre Dresden, Department of Dermatology, University Hospital Carl Gustav Carus, 01307 Dresden, Germany

**Keywords:** metastatic melanoma, targeted therapy, immune checkpoint inhibitor therapy, survival, statutory health insurance data

## Abstract

**Simple Summary:**

Novel therapies have become available in the routine care of metastatic melanoma in recent years. We conducted a retrospective cohort study based on SHI data from Germany (2010–2020) to investigate overall mortality between patients that received different substance classes. We included 463 patients with distant metastases in the main analysis. Classical chemotherapeutics (CTx) as well as targeted therapeutics (TT) and immune checkpoint inhibitors (ICI) showed protective effects after treatment initiation, which decreased over time. Predicted survival of an average patient over five years since first metastasis was best by sequential treatment with ICI and TT. The worst survival was seen in patients treated with TT alone. However, it is conceivable that the observed high survival differences were overestimated due to bias, such as confounding by indication. It is likely that patients treated exclusively with TT were in an extremely serious condition and died before they could have received an ICI.

**Abstract:**

(1) Background: Targeted (TT) and immune checkpoint inhibitor (ICI) therapies have become available in the routine care of metastatic melanoma in recent years. (2) Objective: We compared mortality in patients with metastatic melanoma and different systemic therapies. (3) Methods: A retrospective cohort study, based on pseudonymized health insurance data of about two million individuals from Saxony, Germany, was conducted for the years 2010 to 2020. Only patients with an advanced stage, i.e., distant metastases were considered for the main analysis. Relative survival since metastasis and predicted survivor curves derived from a Cox model were used to assess potential differences in mortality. (4) Results: Relative survival was highest in the subgroup with sequential use of ICI and TT. All treatments except interferon had significant hazard ratios (HR) in the Cox model with time-dependent effects indicating a protective effect after treatment initiation (HR 0.01–0.146) but decreasing over time (HR 1.351–2.310). The predicted survivor curves revealed best survival under ICI-TT treatment and worst survival under TT treatment alone. (5) Conclusions: We found real-world evidence for survival benefits of patients with metastatic melanoma who received sequential ICI and TT treatment. It is conceivable that the observed high survival differences were overestimated due to bias, such as confounding by indication.

## 1. Introduction

In the treatment of metastatic melanoma novel approaches including the use of targeted therapies (TT) and immune checkpoint inhibitors (ICI) have become increasingly available in routine care in recent years [1]. Promising targeted therapies with BRAF (v-raf murine sarcoma viral oncogene homolog B1) inhibitors like vemurafenib and dabrafenib and immune checkpoint inhibitors like nivolumab and pembrolizumab have been approved for treatment of metastatic melanoma in the last decade [2,3,4,5]. The combination of BRAF and MEK (mitogen-activated protein kinase) inhibitors in the context of targeted therapy [6] and the ICI combination nivolumab plus ipilimumab were also approved in Germany during this period. However, the effectiveness of these novel therapies compared to classical chemotherapeutic agents as well as interferon has rarely been investigated outside of clinical trials in Germany [7,8,9]. To find out how the different therapeutic approaches compare in routine care of patients with metastatic melanoma, we aimed at assessing differences in overall survival with TT and ICI compared with chemotherapy and interferon therapy. We also estimated survival effects of the therapies used, taking into account different localizations and the timing (synchronous vs. metachronous) of metastases, as well as sociodemographic variables. Because TT and ICI show very different pharmacological effects and response characteristics, we modeled an interaction between both therapies considering patients, who were treated sequentially with both drugs. The robustness of the results was assessed by sensitivity analyses.

## 2. Materials and Methods

Study population: Pseudonymized routine data from AOK PLUS, a large statutory health insurance in Saxony were used. Pseudonymization comprised the masking or deletion of personal identifiers (exact names were pseudonymized prior to receiving the data; no social security numbers were provided) as well as the generalization of quasi-identifiers (year of birth only, deletion of the last two digits of the zip code, etc.). AOK PLUS covers almost half of the regional population. Approximately two million individuals age ≥18 years from 2010–2020 were available. Periods before 2010 were no longer accessible, as deletion time limits for statutory health insurance data prescribed by German law made this impossible. Age distribution and sex ratio of their beneficiaries in Saxony are comparable to the nationwide population [10]. The data include information from inpatient and outpatient care regarding diagnoses, procedures, and prescriptions (Appendix A), as well as sociodemographic information of the insured such as age and sex.

Case definitions: In agreement with the good practice for secondary data analysis of the German Society for Epidemiology [11], incident melanoma patients were identified by (I) at least an inpatient diagnosis (ICD-10-GM C43) or (II) a combination of specific diagnoses in the outpatient sector (Appendix A). Individuals had to be cancer-free for at least two years (washout period) to be considered newly diagnosed. Thus, patients with a first-time cancer could be estimated for the years 2012 to 2020. Of these, only continuously insured individuals with metastatic melanoma with distant metastases (ICD-10-GM C78 and C79; UICC/AJCC stage IV) were included. As sensitivity analysis, we additionally included persons with locoregional metastases (ICD-10-GM C77; UICC/AJCC stage III) and contrasted the results with the main analysis. The type and temporal occurrence of metastases were mapped individually for each subject. First metastasis within 100 days of initial melanoma diagnosis were defined as synchronous as opposed to metachronous. The time from the appearance of the first metastasis to death, or to the end of the observation period (31 December 2020), was determined for survival analysis. The type of systemic therapy was assessed using various codes from different medical classification systems (Appendix A). This made it possible to distinguish between classical chemotherapeutic agents (CTx) and interferon (IFN), as well as immune checkpoint inhibitors (ICI) and targeted therapies (TT). Only therapies in a period of two years after diagnosis of the first metastasis were considered. The age of each person in years at the time of the first metastasis was determined. In addition, we analyzed the sex distribution. To obtain a proxy for healthy users, we determined the proportion of individuals who had been voluntarily vaccinated against influenza prior to their melanoma diagnosis. We stratified by frequent localizations of metastases and compared the proportion of individuals with different systemic anti-cancer therapies.

Statistical analysis: Relative survival analysis was used to compare survival by therapy, type, and timing of metastasis over a five-year period since first metastasis. Relative survival was estimated according to Perme and Pavlič [12]. Period life tables from Saxony were received on individual request from the State Statistical Office of the Federal State of Saxony. This allowed for estimation of disease-associated excess mortality. Further, a Cox proportional hazards model [13] was used to assess relationships between different treatment regimens and overall survival adjusted for covariates. We used the following variables for modeling: sex, age at first metastasis, metastasis type (six subgroups: bone, lung, liver, brain, bowel, and other sites), timing of first metastasis (synchronous; metachronous), chemotherapy (no; yes), interferon therapy (no; yes), ICI (no; yes), TT (no; yes), and an interaction (sequential usage) of ICI and TT treatment (no; yes). Further we adjusted multivariable models for comorbidities according to Elixhauser [14] and Garland [15], for the last occupational position of the person, for the resection of the primary tumor, for multiple localizations of distant metastases, and for influenza vaccinations prior to the first diagnosis of melanoma. Influenza vaccination was proposed as a proxy for health seeking behavior. In order to make a fair comparison, we weighted the number of vaccinations with the individual risk period of that person (individual time interval between the start of the study and the time of diagnosis of a new-onset melanoma). The reasoning behind this is that people with a higher level of health awareness tend to participate in voluntary vaccinations more than less health-conscious people. Based on our own preliminary work [16] and considerations of [17], we used influenza vaccinations as a proxy for patients’ health behavior. To address possible violations of proportional hazards assumption, we additionally modeled time-varying effects by including interactions between therapy type and the natural logarithm of time since metastasis in years. Based on the Cox model with time-dependent effects, survivor functions were derived to show the absolute predicted survival rates of an average patient with metastatic melanoma for each investigated systemic therapy over a period of 5 years since first metastasis. Statistical analyses were performed in R. We used the relsurv, survival, and survminer packages to calculate relative survival and hazard ratios and to derive the estimated survivor curves from the Cox model. All 95% confidence intervals were calculated throughout the manuscript according to the Wald method [18].

## 3. Results

### 3.1. Descriptive Statistics

Based on our eligibility criteria (Appendix A), we identified 463 patients in our database with newly diagnosed melanoma presented with metastases or developing metastases between 2012 and 2020. The median age at time of first metastasis of the entire cohort was 75 years. Between the subgroups of different metastatic localizations (Table 1) including patients with locoregional metastases from the sensitivity analysis, the median age varied from 71 to 74 years. About 53% of the patients had distant metastases in multiple localizations (Appendix A). The entire cohort and all studied subgroups showed a higher proportion of men (up to 62%). Of 463 patients, 161 received no systemic therapy at all within two years since diagnosis of metastatic disease. Solely 39 patients received chemotherapy, whereas 28 received interferon, 75 received ICI, 16 received TT, and 31 received a sequential therapy with TT and ICI. The remaining 113 patients received a different therapeutic regimen with a combination/sequential use of the aforementioned drug classes (e.g., chemotherapy and ICI; not studied here). The distribution of synchronous and metachronous distant metastases is approximately 50:50 in the overall cohort as well as in the individual subgroups (Table 1). Overall and in all subgroups, the proportion of patients with resected primary tumor was around 90%. Only patients with metastases in liver and lung showed a slightly lower proportion around 87%. 

Documented therapies per patient increased notably over time (2012–2020) for TT, CTx, and especially ICI (Table 2). By contrast, interferon administration for metastatic melanoma patients has decreased since 2018. In detail, the use of immune checkpoint inhibitors increased disproportionately from 2015 to 2020. However, during the same period, the use of TT treatment and the rate of chemotherapy remained almost stable.

### 3.2. Relative Survival

An estimate of survival compared to the general population in Saxony allowed us to contrast cancer-related mortality in different strata. Part A of Figure 1 provides insights into the different dynamics of the various subgroups treated with different systemic therapy regimens. The subgroup without systemic therapy, as well as those treated with classical chemotherapeutic agents had the worst survival. Both had already reached their median survival before one year (50% survival ratio). The same is true for patients treated exclusively with targeted therapeutics (TT). However, compared to the first mentioned, these showed significantly better response rates. In the first six months, the survival in the TT subgroup was the best compared to all investigated therapy subgroups. However, survival decreased dramatically after the first six months and dropped to the lowest level of all after about one year. In contrast, exclusive use of immune checkpoint inhibitors (ICI) or sequential uses of ICI and TT showed the best relative survival ratios. Their median survival was reached as late as one and a half years for ICI-TT, to nearly two years for ICI alone. However, the response of ICI alone was significantly worse than the additional use of TT. After three years, sequential use of ICI and TT showed the best relative survival of all with about 45% survival ratio. Part B of Figure 1 illustrates relative survival depending on the localizations of distant metastases. Six subgroups were contrasted. Relatively common lung as well as brain metastases showed comparable survival ratios of about 25% after three years. Their previous curve progressions were also very similar. While bone metastases showed slightly better survival, patients with liver metastases had the worst outcome of all at 3 years. However, in the long term, in the period greater than 4 years, patients with brain metastases showed the worst survival. The situation was different in patients with metastases in the intestinal tract. In this subgroup, our data showed a survival ratio of over 60% after three years. The residual category, in which all other localizations were lumped, also showed a better survival than patients with bone, lung, brain, and liver metastases, with more than 50% relative survival ratio at 3 years. Within this residual category, there were more codes from ICD-10 C79 than from C78. However, the proportion of patients with multiple localizations of distant metastases (Table 1) was highest in the subgroup with lung metastases (89.5%). Most of the patients in this subgroup had multiple metastases at different localizations and thus theoretically had a worse prognosis than patients with metastases solely in the lungs. Part C of Figure 1 shows the survival difference between synchronous and metachronous metastases in favor of a synchronous diagnosis. Part D gives the relative survival curve of the entire study population (including 95% confidence intervals after Wald). The median survival here was approximately one year and seven months. It should be noted that towards the end of the observation period after 5 years the number of cases became very low in all subgroups.

### 3.3. Cox Regression Analysis

To adjust estimates of therapy effects for potential confounders, we used Cox regression (Table 3). The multivariable Cox model with only time-constant effects revealed statistically significant protective effects only for the ICI therapy (HR = 0.559; 95% CI = (0.412; 0.757)). The point estimates for chemotherapy, interferon, TT, and the interaction of ICI and TT also indicated protective treatment effects but were not significant. Because all therapies violated the proportional hazards assumption, we included time-varying effects for therapy types. This resulted in two coefficients for each type of therapy: one for the effect after one year and another for the change in effect in each subsequent year. Here, chemotherapy, ICI, and TT showed significantly protective effects in the first year, but a significant reduction in effect size in subsequent years. While the time-constant coefficient of TT indicated a hazard ratio below unity (HR = 0.010; 95%-CI = (0.001; 0.081)), the time-dependent coefficient of TT indicated a hazard ratio above unity (HR = 2.310; 95% CI = (1.629; 3.275)). At baseline, TT showed a strong protective effect. However, with increasing time, the protective effect decreased. The same applied in a less pronounced form to chemotherapy and ICI.

### 3.4. Survivor Functions—Predicted Survival of a Hypothetical Average Patient

To illustrate the estimated differences between therapies, we calculated survivor functions (predicted survival estimates) for each therapy type based on the multivariable Cox model with time-dependent effects (Figure 2). The predicted survival curves showed a complex pattern. Predicted survival varied considerably over the observation period of five years. This resulted in overlaps of therapy-specific survival curves. At the start of therapy after diagnosis of the first metastasis, TT and the sequential uses of ICI and TT had the highest predicted survival rates compared with the other therapies or the subgroup without any systemic therapy, indicating a high response rate for these therapy regimes in an average patient. Over time, however, the predicted survival rate of patients with TT decreased most sharply, intersecting with the curve of the subgroup without any documented systemic therapy already after about 600 days. Overall, the subgroup with TT thus showed the worst predicted survival rates. Compared with TT or chemotherapy, patients receiving ICI showed higher predicted survival over the entire five-year period. The sequential uses of ICI and TT showed the best predicted survival over the entire period compared with all other investigated therapy regimes. Note that the order of administration of ICI and TT drugs was not considered here because of the small sample size.

## 4. Discussion

### 4.1. Implementation of Novel Therapies in Routine Care

The implementation of novel systemic therapies for metastatic melanoma in routine care apparently took place. The use of immune checkpoint inhibitors increased disproportionately from 2015 on. This was probably related to the approval of highly effective ICI drugs for metastatic melanoma, such as the anti-PD-1 antibodies nivolumab [19,20,21] or pembrolizumab [22,23] and the ICI combination nivolumab plus ipilimumab. Within the same time span, however, the use of TT treatment stagnated, and the use of interferon therapy declined. However, the rate of chemotherapy remained at a relatively high level even after 2015. It is possible, however, that these classic chemotherapeutic drugs were administered due to a malignant second tumor. It was verified that no other tumor was reported prior to the melanoma diagnosis, but tumors can of course be detected in the follow-up. However, these were not taken into account in the analyses and could therefore be at least partly responsible for the relatively high chemotherapy rate. In addition, we did not have information on the mutation status of the tumor, which can also be important for treatment decision. Patients with BRAF mutation (*BRAF* V600) have a more favorable prognosis than non-mutated patients and may have a treatment advantage by having both ICI and TT as treatment options. Sequential use of ICI and TT was more frequent than TT treatment alone. Current studies investigate which therapy sequence is best for patients with BRAF mutated melanoma. Studies investigating combinations of BRAF/MEK inhibitors and ICI yielded conflicting results [22,24].

### 4.2. Effectiveness of Novel Therapies in Routine Care

The effectiveness of the different types of systemic therapies in routine care was assessed using survival time analyses. Relative and modelled overall survival showed consistent results. Patients treated with an immune checkpoint inhibitor showed a significantly better long-term survival than patients treated with targeted therapies. ICI treatment also had a clear advantage over classical chemotherapeutic agents or no systemic therapies. However, in the first six months, survival of patients treated with ICI was inferior to that of patients treated with TT. These findings are consistent with published results by Ugurel, Röhmel, Ascierto, Flaherty, Grob, Hauschild, Larkin, Long, Lorigan, McArthur, Ribas, Robert, Schadendorf and Garbe [9] based on representative clinical trial data. This observation encouraged clinical studies investigating the sequence and combination of BRAF/MEK inhibitors and anti-PD-1 or anti-PD-L1 antibodies [22,23]. In this context we would like to mention that not only sequencing and combining of ICI and TT appears to be a promising approach but also combining ICI with radiotherapy [25]. Radiation enhances tumor antigen visibility and promotes T cell priming. The combination of radiotherapy and ICI achieves promising survival outcomes and seems to be safe [25]. For example, an analysis of the U. S. National Cancer database detected a significant overall survival benefit of adding stereotactic radiosurgery to ICI in patients with melanoma or non-small cell lung cancer metastatic to the brain [26]. The combination strategy of ICI plus stereotactic radiotherapy is being prospectively investigated in several studies [27,28,29].

However, the survival benefits of patients with specific therapies may have been over-estimated based on our data. In particular, we cannot exclude confounding by indication. One possible explanation is that patients treated exclusively with TT in particular were in an extremely serious condition. These patients may have been treated with a substance from the class of targeted therapeutics in order to achieve the highest possible response. It is conceivable that these patients died before they could have received an immune checkpoint inhibitor subsequently. Thus, the positive effect of the combination of ICI and TT in sequential use was probably overestimated in our analyses since it refers to patients who must have survived long enough to receive both therapies. Some of the patients treated exclusively with TT should actually be included in the subgroup of those treated with ICI and TT. Thus, the actual possible survival benefit from ICI-TT treatment would be lower and would lie between the curves of TT and ICI-TT. Even though we could not completely mitigate the bias due to confounding by indication, we aimed to adjust for confounding by indication by fitting our models with covariates, including socio-demographic characteristics. For example, the status of the primary tumor resection was assessed. Presumably, patients without resection had a very serious disease. Furthermore, the subgroup treated exclusively with TT was very small, which induces low precision of statistical estimates, too.

The total cohort from Saxony analyzed here showed a median survival about 7 months longer compared to an estimate for metastatic melanoma patients from 2000–2016 in eastern Germany, which was based on cancer registry data [7]. However, since these are different databases, one should interpret this improvement over time with caution.

Almost 35% of the included patients had no documented systemic therapy within two years of diagnosis of first metastasis. This may have created a bias in the survival analyses by including patients in the reference group who were incorrectly considered as not receiving treatment. Moreover, these patients may have been treated surgically with subsequent tumor follow-up in a curative setting or received best supportive care in a palliative treatment setting.

Our observation that survival may vary substantially by localization of distant metastasis should be investigated more deeply with a larger database. The worst 3-year survival had patients with liver metastases, corresponding to ICD-10-GM code C78.7, compared to other localizations. However, after three years, patients with lung metastases had only a slightly higher survival advantage than patients with liver metastases. Patients with brain metastases had a comparably poor survival. However, patients with lung metastases frequently had second metastases in different localizations (89.5%). Brain metastases and lung metastases together, for example, were particularly common (Table 1). In general, patients with multiple localizations of distant metastases have a worse prognosis than those with only one localization. The high proportion of multiple localizations may be the reason why patients with lung metastases showed such poor survival rates in our analysis. Patients with intestinal metastases showed the best survival of all subgroups studied. However, this localization is rather rare. Kaplan–Meier survival curves for death in cancers of unknown primary (CUP), depending on localization of metastases from [30] showed qualitatively the same pattern as our analysis with liver metastases being associated with worst and gastro-intestinal metastases being associated with best survival prognosis, while brain and lung metastases were in between. Unfortunately, there was no information available on the size and number of metastatic lesions per localization. In addition, we cannot exclude the possibility that the localizations of the metastases were partially assigned wrong, e.g., due to incorrect coding, and our results are therefore biased.

## 5. Conclusions

In recent years, we have seen an increase in novel therapies in routine care. In particular, immune checkpoint inhibitors were increasingly used since 2015. We found real-world evidence of survival benefits in patients with metastatic melanoma treated sequentially with ICI and TT. However, it is likely that the observed survival differences were affected by bias, such as confounding by indication, and that we had therefore overestimated the survival benefits.

## Figures and Tables

**Figure 1 cancers-13-06150-f001:**
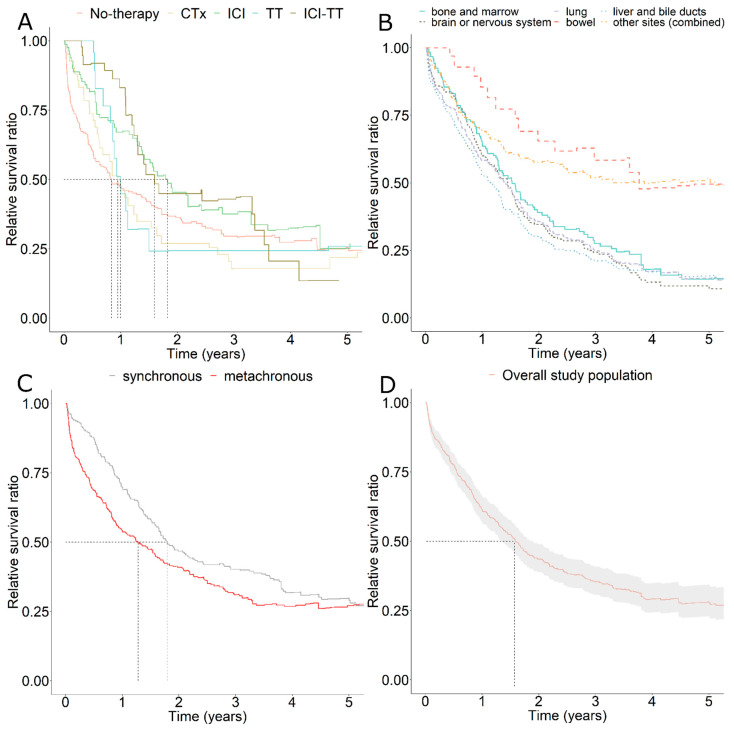
Relative survival of metastatic melanoma patients since first metastasis compared to the background mortality of all Saxon residents (**A**) depending on the therapy received. The subgroups with certain therapies were exclusive and no other combinations except ICI and TT were displayed. CTx: chemotherapy, ICI: Immune Checkpoint Inhibitor therapy, TT: Targeted therapy, ICI-TT, ICI, and TT in sequential use (order not considered); Relative survival (**B**) depending on the localization of distant metastases (five common sites and a sixth subgroup with the remaining localizations of metastases) or (**C**) depending on the time of occurrence of the metastases (synchronous vs. metachronous). (**D**) Relative survival of the total cohort studied. Risk tables for subgroups and time points (yearwise) were given respectively below each graph. Median survival time for each curve were indicated with a dashed line.

**Figure 2 cancers-13-06150-f002:**
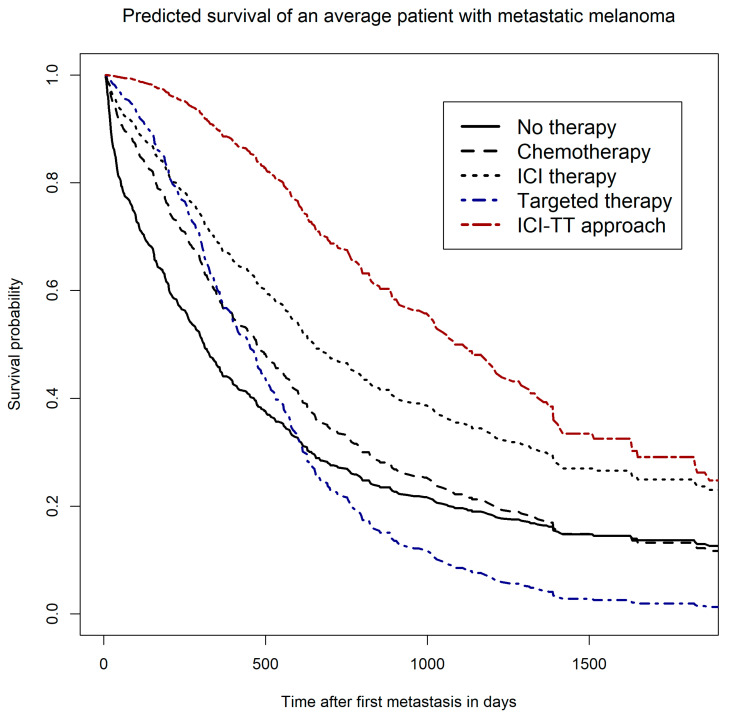
Predicted survival of an average patient (mean adjusted) since first metastasis for chemotherapy, immune checkpoint inhibitor therapy (ICI), targeted therapy (TT), sequential use of ICI and TT, and no documented systemic therapy.

**Table 1 cancers-13-06150-t001:** Information on study cohort and subgroups with different types of metastasis.

Main Analysis											Sensitivity Analysis	
Population	Overall Sample		Bone C79.5 *		Brain C79.3 and C79.4 *		Liver C78.7 *		Lung C78.0 *		Locoregional Metastasis	
*n* per group	463		110		150		170		220		588	
Predictor	*n*	(Q1; Q3)/%	*n*	(Q1; Q3)/%	*n*	(Q1; Q3)/%	*n*	(Q1; Q3)/%	*n*	(Q1; Q3)/%	*n*	(Q1; Q3)/%
Median age at first metastasis	75	(64; 82)	70	(56; 80)	71	(57; 80)	73	(62; 80)	74	(64; 81)	74	(63; 82)
Sex												
Female	209	45.1	48	43.6	57	38.0	71	41.8	93	42.3	270	45.9
Male	254	54.9	62	56.4	93	62.0	99	58.2	127	57.7	318	54.1
Type of systemic therapy												
No systemic therapy	161	34.8	28	25.5	40	26.7	37	21.8	49	22.3	223	37.9
Chemotherapy	125	27.0	38	34.5	48	32.0	63	37.1	79	35.9	145	24.7
Chemotherapy exclusive	39	8.4	5	4.5	8	5.3	19	11.2	25	11.4	45	7.7
Interferon therapy	79	17.1	21	19.1	30	20.0	30	17.6	42	19.1	107	18.2
Interferon therapy exclusive	28	6.0	2	1.8	5	3.3	5	2.9	8	3.6	46	7.8
Immune checkpoint inhibitor therapy (ICI)	203	43.8	67	60.9	76	50.7	94	55.3	122	55.5	237	40.3
ICI exclusive	75	16.2	19	17.3	20	13.3	29	17.1	42	19.1	87	14.8
Targeted therapy (TT)	79	17.1	26	23.6	46	30.7	41	24.1	45	20.5	85	14.5
TT exclusive	16	3.5	4	3.6	9	6.0	8	4.7	8	3.6	20	3.4
ICI + TT sequential	50	10.8	18	16.4	27	18.0	27	15.9	30	13.6	51	8.7
ICI + TT exclusive	31	6.7	11	10.0	17	11.3	16	9.4	18	8.2	31	5.3
Localization of distant metastases (multiple localizations per patient are counted)												
C78	300	64.8	79	71.8	110	73.3	170	100.0	220	100.0	300	51.0
C79	377	81.4	110	100.0	150	100.0	123	72.4	179	81.4	377	64.1
Lung C78.0 *	220	47.5	68	61.8	92	61.3	102	60.0	220	100.0	220	37.4
Brain or nervous system C79.3 C79.4 *	150	32.4	46	41.8	150	100.0	59	34.7	92	41.8	150	25.5
Bowel C78.4 C78.5 *	23	5.0	12	10.9	11	7.3	14	8.2	16	7.3	23	3.9
Liver and bile ducts C78.7 *	170	36.7	58	52.7	59	39.3	170	100.0	102	46.4	171	29.1
Bone and marrow C79.5 *	110	23.8	110	100.0	46	30.7	58	34.1	68	30.9	110	18.7
Other sites (combined)	182	39.3	110	100.0	2	1.3	37	21.8	54	24.5	182	31.0
Number of patients with multiple localizations	248	53.6	88	80.0	119	79.3	141	82.9	197	89.5	248	42.2
Metastasis timing												
Synchronous	215	46.4	56	50.9	78	52.0	89	52.4	107	48.6	287	48.8
Metachronous	248	53.6	54	49.1	72	48.0	81	47.6	113	51.4	301	51.2
Other												
Resection of primary tumor	417	90.1	100	90.9	136	90.7	147	86.5	193	87.7	534	90.8
Influenza vaccination prior diagnosis	280	60.5	60	54.5	77	51.3	96	56.5	131	59.5	366	62.2

*n* = number of observations; Q1 first quantile; Q3 third quantile; percent values refer to the number of samples per group; * ICD-10-GM, International Classification of Diseases, 10th Revision, German Modification; C77 locoregional metastasis, C78 & C79 distant metastasis.

**Table 2 cancers-13-06150-t002:** Therapy types per patient documented over time during the observation period 2012 to 2020 for chemotherapy (CTx), immune checkpoint inhibitor therapy (ICI), targeted therapy (TT), and interferon therapy (IFN). Number of patients: *n* = 588 (including persons with locoregional metastases).

		Years		
		2012–2014	2015–2017	2018–2020
Number of patients in therapy	CTx	56	97	81
	IFN	92	90	18
	ICI	36	157	282
	TT	31	46	43

CTx chemotherapy, IFN interferon therapy, ICI immune checkpoint inhibitor therapy, TT targeted therapy.

**Table 3 cancers-13-06150-t003:** Univariable and multivariable Cox regression models for time since first metastasis. Multivariable models were adjusted for age, sex, type and timing of metastasis, comorbidities, occupational position, occurrence of multiple localizations of distant metastases, and influenza vaccinations prior diagnosis of melanoma weighted for personal time under risk. Because the systemic therapies showed a violation of the proportional hazard assumption, time-varying effects were included in the cox model (log function was used). This allowed the survival rates of the therapies to intersect over time.

	Main Analysis				Sensitivity Analysis ^s^
Number of Included Persons	N = 463				N = 588
				Hazard Ratio (95%-CI) (Multivariable ^+^, Time-Constant Effects, Time-Dependent Effects)	Hazard Ratio (95%-CI) (Multivariable ^+^, Time-Constant Effects, Time-Dependent Effects)
			Hazard Ratio (95%-CI) (Multivariable ^+^, Time-Constant Effects)		
		Hazard Ratio (95%-CI) (Univariable)			
Predictor	Reference				
Age at metastasis (49–75 yrs)	23–49 yrs	1.582 (0.965; 2.594)	1.680 (0.951; 2.970)	1.802 (1.008; 3.223)	1.993 (1.118; 3.550)
Age at metastasis (75–100 yrs)	23–49 yrs	2.589 (1.592; 4.210)	2.555 (1.344; 4.858)	2.645 (1.375; 5.087)	3.019 (1.585; 5.752)
Female sex	Male sex	0.872 (0.703; 1.081)	0.960 (0.745; 1.238)	0.947 (0.734; 1.222)	0.945 (0.742; 1.204)
Metachronous Metastasis	Synchronous M.	1.335 (1.077; 1.655)	1.160 (0.897; 1.500)	1.207 (0.932; 1.562)	1.253 (0.987; 1.590)
Type of systemic therapy					
Chemotherapy (CTx)	No chemotherapy	1.077 (0.852; 1.362)	0.921 (0.705; 1.204)	0.146 (0.041; 0.517)	0.160 (0.047; 0.542)
Interaction chemotherapy with time since metastasis	-	-	-	1.382 (1.114; 1.715)	1.365 (1.110; 1.678)
Interferon therapy	No Interferon	0.502 (0.369; 0.682)	0.751 (0.530; 1.065)	0.874 (0.176; 4.328)	1.144 (0.252; 5.200)
Interaction interferon therapy with time since metastasis	-	-	-	0.958 (0.736; 1.247)	0.890 (0.694; 1.143)
Targeted therapy (TT)	No TT	0.934 (0.710; 1.228)	0.990 (0.621; 1.578)	0.010 (0.001; 0.081)	0.018 (0.003; 0.123)
Interaction targeted therapy with time since metastasis	-	-	-	2.310 (1.629; 3.275)	2.132 (1.543; 2.947)
Immune checkpoint inhibitor therapy (ICI)	No ICI	0.706 (0.567; 0.877)	0.559 (0.412; 0.757)	0.113 (0.035; 0.368)	0.092 (0.029; 0.294)
Interaction ICI with time since metastasis	-	-	-	1.351 (1.097; 1.665)	1.420 (1.162; 1.737)
Interaction ICI with TT	-	0.789 (0.560; 1.110)	0.705 (0.378; 1.313)	0.394 (0.201; 0.775)	0.356 (0.184; 0.688)
Localization of distant metastases *					
Lung C78.0	No lung	1.609 (1.296; 1.997)	1.207 (0.838; 1.739)	1.311 (0.909; 1.891)	1.692 (1.190; 2.406)
Brain or nervous system C79.3 C79.4	No brain	1.442 (1.158; 1.795)	1.174 (0.824; 1.672)	1.182 (0.833; 1.676)	1.755 (1.280; 2.406)
Bowel C78.4 C78.5	no bowel	0.511 (0.294; 0.891)	0.438 (0.241; 0.795)	0.431 (0.235; 0.788)	0.425 (0.230; 0.785)
Liver and bile ducts C78.7	no liver	1.673 (1.348; 2.077)	1.558 (1.156; 2.101)	1.525 (1.131; 2.058)	1.917 (1.434; 2.562)
Bone and marrow C79.5	no bone	1.200 (0.943; 1.527)	1.030 (0.754; 1.406)	1.069 (0.782; 1.460)	1.383 (1.034; 1.849)
Other sites (combined)	no lung/brain/bowel/liver/bone	0.515 (0.408; 0.651)	0.607 (0.412; 0.894)	0.650 (0.442; 0.956)	1.045 (0.758; 1.441)


^+^ multivariable models were further adjusted for 21 selected Elixhauser comorbidities, last occupational position of the person, resection of the primary tumor, number of influenza vaccinations divided by individual risk period before first diagnosis of melanoma in years and an indicator for multiple localizations of distant metastases (not shown); time since first metastasis in years; * ICD-10-GM codes for secondary neoplasms; ^S^ for sensitivity analysis we included all patients with only locoregional metastases.

## Data Availability

The data used are routine health insurance data from AOK PLUS, which are confidential and therefore not publicly available. The study is supported by AOK PLUS, with whom the Center for Evidence-based Healthcare has a Data Use and Transfer Agreement. Personal data of the beneficiaries were pseudonymized through AOK PLUS before data sharing. Personal identifiers were masked or deleted (exact names were pseudonymized prior to receiving the data; no social security numbers were provided). Quasi-identifiers were generalized (year of birth only, deletion of the last two digits of the zip code, etc.). The processing and analysis of sensitive data took place exclusively on the specially protected servers at Dresden University Hospital. Results of data analyses were shared in aggregate form only.

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
