# Peer review of "Implementation and Effectiveness of Novel Therapeutic Substances for Advanced Malignant Melanoma in Saxony, Germany, 2010–2020—Cohort Study Based on Administrative Data"

_cancers, 2021, doi:10.3390/cancers13246150_

Round 1

Reviewer 1 Report

The authors present a paper about "Implementation and Effectiveness of Novel Therapeutic Substances for Advanced Malignant Melanoma".

The topic is absolutely interesting and the overall quality of the manuscript is of high level.

The researchers have conducted an extensive analysis of data on a considerable number of patients.

I have a few suggestions which could help the readers to undertand more easily the content of the paper as follows:

1) The authors state they used "pseudonymized health insurance data of about 2 million": could the more clearly explain the concept of "pseudonymized"? Was an explict consent provided to patients to use their data?

2) The range of accrual is considerable (2010-2020) however in my view the authors should explain the reason for such choise: why did not they choose another accrual period with a more mature follow-up time (such as for instance 2008-2018?)

3) The main result of the study is absolutely interestinf because "Relative survival was highest in the subgroup with sequential use of ICI and TT". Recently the concepts of Peri-Induction Radiotherapy (PIR) and Post-Escape Radiotherapy (PER) have been highlighted in literature (see PMID: 33847208 for furhte details). In my opinion in the discussion section it would be an addition to consider the valuable role of combination therapy between ICI and local therapies such as radiotherapy.

4) I have some concerns about the possible overestimation of the survival benefits (as correclty underlined by the authors in the conclusion) but I believe such point should be addressed more clearly also in the discussion.

Reviewer 2 Report

This is a clear and concise presentation of a complex analysis that should be of value to clinicians treating individuals with advanced melanoma. Almost all questions I had while reading were addressed in the discussion, such as bias with regard to treatment modality introduced by the tumor characteristics (presence of a targetable mutation) or co-morbidities of the patient at diagnosis that preclude some therapies.

I have two requests for clarification:

Line 113: Is "individual risk period" the same as age at time of diagnosis? 

Table I: Would it be possible to create a supplementary figure to illustrate the relative numbers of metastatic lesion combinations? It was difficult to derive how many individuals had a metastatic lesion in only one location as opposed to multiple locations from Table I. 

Minor edits:

  • Localization Influenza is spelled incorrectly in Table I, last line.
  • Inconsistent spelling of localisation/localization (see Table I)

Reviewer 3 Report

The submitted manuscript is sounding and interesting.

Data analysis and statistical methods are adequate and results appear unbiased.

I suggest only a text revision for typos and an addition at lines 111-114: authors should better explain the reason of choice of influenza vaccination in this context and add at least one reference about this.
